# BENFORD-QUANT: A BENFORD'S LAW-INSPIRED NON-UNIFORM QUANTIZER FOR EFFICIENT LANGUAGE MODELS

## ABSTRACT

The rapid growth of Large Language Models (LLMs) intensifies the need for effective compression, with weight quantization being the most widely adopted technique. Standard uniform quantizers assume that parameters are evenly distributed, an assumption at odds with the highly skewed distributions observed in practice. We propose **Benford-Quant** (BENQ), a simple, data-free non-uniform quantizer inspired by Benford's Law, which predicts that leading digits follow a logarithmic distribution. BENQ replaces the uniform grid with a log-spaced codebook, dedicating more resolution to the frequent small-magnitude weights. We provide both theoretical intuition and empirical evidence: (i) weights in transformer *transformational* layers adhere closely to Benford statistics, while normalization layers systematically deviate; (ii) on Small Language Models (SLMs), BENQ consistently improves perplexity, reducing 4-bit perplexity on Gemma-270M by more than 10%; and (iii) on larger LLMs, it remains competitive, with differences explained by over-parameterization effects. Our results indicate that incorporating a Benford-inspired prior into quantization grids is a low-cost modification that yields accuracy gains in aggressive few-bit regimes. Although it is not able to surpass the state of the art in tasks such as perplexity and LAMBADA, the BENQ approach can be hybridized with other quantization methods—such as SmoothQuant and Activation-Aware Quantization—without major pipeline modification, potentially improving their performance.

## 1 INTRODUCTION

Large Language Models (LLMs) deliver state-of-the-art results across NLP tasks, yet their memory and latency footprints hinder broad deployment (Touvron et al., 2023; Jiang et al., 2023). Post-training quantization (PTQ) is a practical remedy: by mapping full-precision weights to few-bit integers, it compresses models and often accelerates inference with modest accuracy loss (Frantar et al., 2022; Dettmers et al., 2023). The de-facto baseline, round-to-nearest (RTN) on a *uniform* grid, is simple and hardware-friendly—but it implicitly assumes that parameters occupy the dynamic range evenly.

Empirically, neural weights are highly non-uniform and concentrate near zero (Han et al., 2015). In low-bit regimes (e.g., 3–4 bits), uniform grids spend disproportionate capacity on rare large magnitudes while under-resolving dense near-zero regions; the mismatch is exacerbated in layers whose weight magnitudes span multiple decades. This has spurred a broad literature on *non-uniform* or *distribution-aware* quantization, from classic logarithmic level schedules (Miyashita et al., 2016) to modern per-layer, learned or optimized codebooks for LLMs (Zhao & Yuan, 2025). In parallel, activation-aware schemes (e.g., AWQ (Lin et al., 2024), SmoothQuant (Lin et al., 2024)) reduce sensitivity to outliers and can be combined with weight-only PTQ (Lin et al., 2024; Xiao et al., 2023).

We revisit a classic regularity of natural data—*Benford's Law* (Benford, 1938)—and show that many *transformational* layers in modern transformers (linear/attention/FFN) exhibit *Benford-like* leading-digit statistics, whereas normalization layers systematically do not, as illustrated in Figure 1b. Beyond empirical evidence, we give a log-domain rationale: multiplicative stochastic optimization (SGD with decay and adaptive preconditioning) induces broad mixtures in $\log |w|$, yielding near-uniform mantissas and thus Benford-like behavior. Notably, prior work has leveraged Benford's Law as an

*analysis or training signal*: Sahu et al. (2021) propose a model as a predictor of generalization and a validation-free early-stopping criterion, while Ott et al. (2025a) regularize significant-digit histograms to improve generalization in low-data regimes.

In this work, we propose BENQ, a *data-free* non-uniform quantizer that replaces the linear codebook with a *log-spaced* grid derived from a Benford-inspired prior, and applies it *selectively* to transformational layers while leaving stability-critical parameters (e.g., LayerNorm scales, embeddings) in higher precision. Conceptually, BENQ allocates more resolution to the statistically prevalent small-magnitude weights, aligning representational capacity with observed weight statistics rather than with a uniform dynamic range. To guide the development of this work, the following research questions were established:

**RQ1: Benford's Law Compliance.** Do the parameters and activations of large language models adhere to Benford's Law?

**RQ2: Benford-Quant Efficacy.** Can a non-uniform, Benford-inspired quantization scheme outperform standard uniform methods?

**RQ3: Performance Benchmarks.** What is the performance (e.g., perplexity) of a model quantized with BENQ compared to other methods like RTN or GPTQ?

The experiments revealed that, on *Small Language Models* (SLMs), BENQ consistently improves perplexity over uniform RTN at 3–4 bits (e.g., Gemma-3-270M: 32.3 vs. 39.5 at 4b; Gemma-3-1B-it: 31.9 vs. 32.3 at 4b). On mid-large-sized LLMs (e.g., OPT-1.3B, BLOOM-1b1/1b7, OPT-66B, Qwen-72B), BENQ remains competitive compared to uniform RTN or GPTQ. We analyze when and why the Benford prior helps (distributional shape, layer role) and where it saturates (flattened spectra in larger models), and we ablate level spacing to show the advantage of Benford/log grids over naive non-uniform alternatives. Compared to learned/optimized non-uniform schemes (Zhao & Yuan, 2025), BENQ trades per-layer search for a *principled, fixed geometry* grounded in a well-studied statistical law, avoiding heavy calibration or optimization.

Crucially, BENQ targets *weight* quantization and is orthogonal to activation-aware or calibration-heavy methods (e.g., AWQ (Lin et al., 2024), SmoothQuant (Lin et al., 2024)) and second-order/PTQ optimizers (e.g., GPTQ). This makes BENQ a *drop-in* codebook amenable to hybridization: it can replace uniform grids inside group-wise PTQ, complement activation smoothing, or initialize codebooks for optimized PTQ.

**Contributions.**

- **Benford link & layer-wise dichotomy.** We document that transformer *transformational* weights exhibit Benford-like leading-digit statistics, whereas normalization layers do not; we support this with a log-domain rationale based on multiplicative training dynamics (Section 2).

- **A simple, selective non-uniform quantizer.** BENQ uses a Benford-inspired, log-spaced codebook and a selective policy; it is data-free, hardware-friendly, and drop-in.

- **Evidence across scales & positioning.** BENQ yields consistent gains over uniform RTN on SLMs and remains competitive on mid-large-sized LLMs; ablations confirm that Benford/log spacing matters beyond "any" non-uniform grid. Unlike prior Benford-based *training* signals (Sahu et al., 2021; Ott et al., 2025a), our approach uses Benford as a *quantization prior*, enabling post-training compression without retraining. Relative to general non-uniform quantization (Miyashita et al., 2016; Zhao & Yuan, 2025), we provide a principled, calibration-light alternative derived from a pre-defined statistical law.

Overall, our results indicate that incorporating a Benford-informed prior into the codebook design provides a low-overhead approach to recovering accuracy in aggressive few-bit regimes, particularly for models whose weight magnitudes span multiple orders of magnitude, and natural building block for hybrid quantization pipelines.

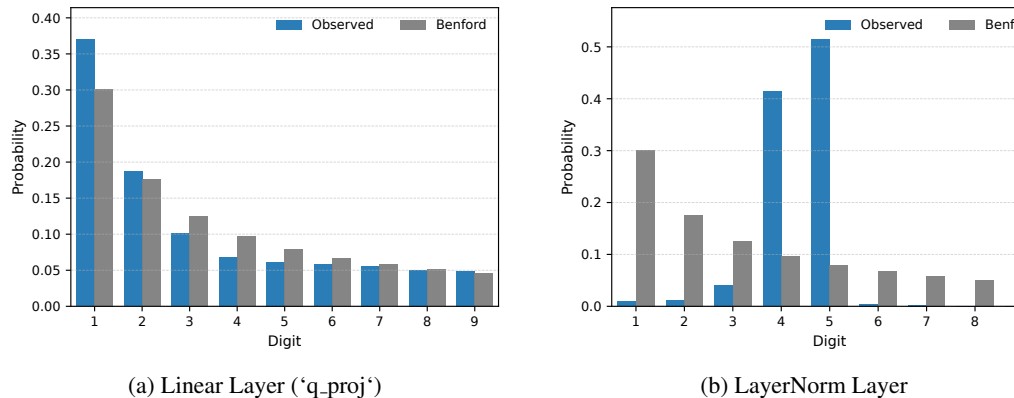

(a) Linear Layer ('q_proj')  (b) LayerNorm Layer

Figure 1: The dichotomy of Benford's Law compliance in Llama-3-8B. (a) The weights of a transformational linear layer strongly adhere to the Benford distribution. (b) In contrast, the weights of a 'LayerNorm' layer systematically violate the law, with their first digits overwhelmingly concentrated on a single value. **Conclusion:** This analysis provides the empirical motivation for both our log-uniform grid and our selective quantization strategy.

## 2 WHY DO WE EXPECT BENFORD-LIKE WEIGHTS?

**Benford preliminaries.** Let $S_{10}(x) \in [1, 10)$ be the base-10 significand, $x = S_{10}(x) \cdot 10^k$. Benford's Law gives

$$\mathbb{P}(\lfloor S_{10}(x) \rfloor = d) = \log_{10}\big(1 + \tfrac{1}{d}\big), \quad d = 1, \dots, 9. \tag{1}$$

A standard characterization states: $x$ is Benford $\Leftrightarrow$ the fractional part of $\log_{10} |x|$ is uniform on $[0, 1)$ (Hill, 1995b). Thus, Benford-like behavior reduces to *equidistribution modulo* 1 in the log domain.

**Multiplicative training dynamics.** For a scalar weight $w_t$ in a linear/affine map, a broad class of optimizers admits

$$w_{t+1} = (1 - \eta_t \lambda) w_t - \eta_t \phi_t g_t = M_t w_t + \varepsilon_t, \tag{2}$$

where $\lambda \geq 0$ (decoupled weight decay), $\eta_t$ is the step size, $\phi_t > 0$ a preconditioner (e.g., Adam), $g_t$ a stochastic gradient, and $\varepsilon_t$ an additive residual. When $|M_t w_t|$ dominates $|\varepsilon_t|$ during nontrivial epochs, the magnitude evolves approximately multiplicatively:

$$\log |w_{t+1}| \approx \log |w_t| + \log |M_t|. \tag{3}$$

Random fluctuations in $\eta_t, \phi_t$ and data/curvature induce a noisy random walk in $\log |w_t|$, producing broad log-distributions (often close to lognormal). Aggregating across layers and training phases yields mixtures of such log-broad components.

**Matrix products and Benford.** Beyond temporal evolution, the *spatial structure* of neural networks also promotes Benford-like behavior. Each forward pass involves repeated matrix–vector multiplications,

$$\mathbf{h}_{\ell+1} = \mathbf{W}_\ell \mathbf{h}_\ell, \tag{4}$$

so entries of $\mathbf{h}_{\ell+1}$ are sums of products of the form $\prod_{j=1}^{k} w_j h_j$. Classical results (e.g., Hill 1995a;c) show that products of independent, non-degenerate random variables tend to produce significands that are uniformly distributed in the log domain, hence Benford. In deep networks, activations at later layers accumulate multiplicative contributions from many random weights, further broadening the log-distribution of effective coefficients. This complementary perspective explains why even static weight matrices (not only their SGD trajectories) naturally exhibit Benford-like first-digit histograms.

**Consequence: near-uniform log mantissas.** Broad, continuous log-distributions and random mixtures are classical mechanisms leading to equidistribution of $\{\log_{10} |w|\}$ modulo 1, hence Benford-like leading digits. This rationale applies to *transformational* weights (linear/attention/FFN). Parameters tightly anchored to a narrow scale—e.g., LayerNorm scales acting as learned damping factors—*violate* the broadness condition and need not be Benford, matching our empirical dichotomy.

**Implication for quantization.** If $\{\log_{10}|w|\}$ is near-uniform, mass is spread across decades, with highest density near zero. A *log-spaced* grid allocates more levels where weights concentrate, reducing expected distortion at fixed bit width. Conversely, for narrow-scale parameters (e.g., LayerNorm), log spacing is suboptimal, motivating *selective* application.

*Note.* Benford compliance (uniform mantissa in $\log_{10}$) does not imply a strictly log-uniform density; our grid is a practical proxy that captures the near-zero concentration that matters for quantization.

## 3 PROBLEM SETTING

The rationale in Section 2 motivates non-uniform, log-spaced grids for transformational weights and a selective policy for stability-critical parameters.

Post-training quantization maps a full-precision tensor $\mathbf{W} \in \mathbb{R}^{m \times n}$ to low-bit integers $\mathbf{W}_q$ via a quantizer $Q(\cdot)$ and dequantizer $DQ(\cdot)$, minimizing $\|\mathbf{W} - DQ(Q(\mathbf{W}))\|$.

**Uniform RTN (baseline).** A $B$-bit symmetric uniform quantizer uses $2^B$ evenly spaced levels with scale $s$:

$$\mathbf{W}_q = Q(\mathbf{W}) = \min \left( \max \left( \text{round}(\mathbf{W}/s), -2^{B-1} \right), 2^{B-1} - 1 \right), \quad DQ(\mathbf{W}_q) = \mathbf{W}_q \cdot s. \quad (5)$$

Choosing $s$ by $\max|\mathbf{W}|/(2^{B-1} - 1)$ is outlier-sensitive; group-wise scaling mitigates this by partitioning $\mathbf{W}$ and using per-group scales (Dettmers et al., 2023).

**Goal.** Design a non-uniform set of levels $\mathcal{L} = \{l_1, \ldots, l_{2^B}\}$ that better matches the empirical log-scale structure of weights, thereby reducing distortion at fixed $B$. In our case, $\mathcal{L}$ is log-spaced and applied selectively (Section 5.1); the algorithmic details follow in Section 4.

## 4 THE BENFORD-QUANT METHOD

Benford-Quant is a post-training, data-free quantization method designed to align the quantization grid with the empirically observed logarithmic distribution of transformer weights. The method consists of three core components: (1) a distributional prior based on Benford's Law, (2) a group-wise quantization algorithm that maps weights to a non-uniform, log-spaced grid, and (3) a selective application strategy that targets only transformational layers.

**Benford's Law as a Distributional Prior.** Benford's Law (Benford, 1938) states that the probability of a number having a first significant digit $d \in \{1, \ldots, 9\}$ is given by:

$$P(d) = \log_{10} \left( 1 + \frac{1}{d} \right). \quad (6)$$

This distribution arises from processes involving scale invariance and implies that values are distributed logarithmically across orders of magnitude. We leverage this principle as a strong prior for the distribution of neural network weights, using it to inform the geometry of our quantization grid.

**Log-Uniform Quantization Grid.** To match the prior from Equation (6), we construct a non-uniform set of $2^B$ quantization levels, $\mathcal{L}$, that are spaced logarithmically. For symmetric quantization with $B$ bits, we first generate the $2^{B-1}$ positive levels, $\mathcal{L}^+$, within the normalized range $(0, 1]$. These levels are spaced evenly in the log domain, concentrating them near zero:

$$\mathcal{L}^+ = \left\{ \exp(x) \mid x \in \left\{ \log(\epsilon) + i \cdot \frac{\log(1.0) - \log(\epsilon)}{2^{B-1} - 1} \,\middle|\, i = 0, 1, \ldots, 2^{B-1} - 1 \right\} \right\} \quad (7)$$

where $\epsilon$ is a small constant (e.g., $10^{-7}$) to avoid numerical instability. The full, symmetric grid $\mathcal{L}$ is then constructed as the union of the positive levels and their negations, $\mathcal{L} = \mathcal{L}^+ \cup (-\mathcal{L}^+)$. This design inherently allocates more representational capacity to the more frequent low-magnitude weights.

**The Quantization and Dequantization Procedure.** The core procedure applies this non-uniform grid to a weight tensor $\mathbf{W}$ in a group-wise fashion. The full process is detailed in Algorithm 1. For each block of weights $\mathbf{w}_g$ of size $G$, we first compute a scale $s_g = \max(|\mathbf{w}_g|)$ to normalize the block to $[-1, 1]$. Then, each normalized weight is mapped to the index of the closest level in our static grid $\mathcal{L}$.

---

**Algorithm 1** The Benford-Quant Quantization Procedure

---

**Require:** Weight tensor $\mathbf{W}$, bit-width $B$, group size $G$.
**Ensure:** Quantized indices $\mathbf{W}_q$, scales $\mathbf{S}$.
1: $\mathcal{L} \leftarrow$ GenerateLogUniformLevels($B$)   ▷ Pre-compute the $2^B$ non-uniform levels in $[-1, 1]$
2: $\mathbf{W}' \leftarrow$ reshape($\mathbf{W}, (-1, G)$)   ▷ Reshape W into blocks of size G
3: Initialize empty tensors $\mathbf{W}_q$ and $\mathbf{S}$ for outputs.
4: **for** each block $\mathbf{w}_g$ in $\mathbf{W}'$ **do**
5:    $s_g \leftarrow \max(|\mathbf{w}_g|)$   ▷ Compute the block's scale
6:    $\hat{\mathbf{w}}_g \leftarrow \mathbf{w}_g / s_g$   ▷ Normalize block to $[-1, 1]$
7:       ▷ Find index of nearest level for all values in the block (vectorized)
8:    $\mathbf{i}_g \leftarrow \arg\min_{j \in \{1,...,2^B\}} |\hat{\mathbf{w}}_g^{\text{unsqueeze}} - \mathcal{L}_j|$
9:    Append $\mathbf{i}_g$ to $\mathbf{W}_q$; Append $s_g$ to $\mathbf{S}$
10: **end for**
11: **return** $\mathbf{W}_q, \mathbf{S}$

---

The dequantization process is a simple reversal. Given the integer indices $\mathbf{W}_q$ and the scales $\mathbf{S}$, the reconstructed weight tensor $\tilde{\mathbf{W}}$ is obtained by first performing a lookup into the level grid and then rescaling each block:

$$\tilde{\mathbf{w}}_g = \mathcal{L}[\mathbf{i}_g] \cdot s_g \tag{8}$$

where $\mathcal{L}[\mathbf{i}_g]$ denotes the element-wise lookup operation for the indices corresponding to block $g$. In the appendix we present a practical example of the method (Figure 5)

**Selective Quantization Strategy.** Our empirical findings in Section 5.1 reveal that 'LayerNorm' weights do not follow the logarithmic distribution assumed by our method. Applying a log-uniform quantizer to their tightly clustered, near-constant distributions is theoretically and practically suboptimal. We therefore adopt a selective quantization strategy: only 'nn.Linear' layers are quantized. Critical stability layers, such as 'nn.LayerNorm' and token 'nn.Embedding' layers, are maintained in their native FP16 precision. This surgical approach preserves model stability with a negligible memory overhead, as these layers constitute a tiny fraction of the total model parameters.

## 5 EXPERIMENTS AND RESULTS

Our experiments are designed to answer our core research questions. We evaluate on several transformer-based model families: Gemma3, Qwen, Llama, OPT and Bloom (Team et al., 2024; Bai et al., 2023; Grattafiori et al., 2024; Zhang et al., 2022; Workshop et al., 2022). All perplexity (defined as $2^{H(p)}$, where $H(p)$ is the entropy of the model's prediction) evaluations are conducted on the test split of WikiText-2 (Merity et al., 2016) on a single computer equipped with an AMD Ryzen Threadripper 7960X 24-Cores @ 5360MHz, 256 GB of DDR5 RAM and a NVIDIA H200 GPU.

### 5.1 RQ1: INVESTIGATING BENFORD'S LAW IN TRANSFORMERS

**Setup.** To establish the foundation for our method, we first analyze the distribution of the first significant digit for every parameter tensor in our test models. We compare the observed distribution against the theoretical Benford distribution using Mean Absolute Deviation (MAD) metric, defined by Cerqueti & Lupi (2023) as

$$MAD = \frac{1}{9} \sum_{i=1}^{9} |p_i - b_i|, \tag{9}$$

where $p_i$ denotes the probability of digit $i$ empirically observed, and $b_i$ the corresponding Benford-expected probability.

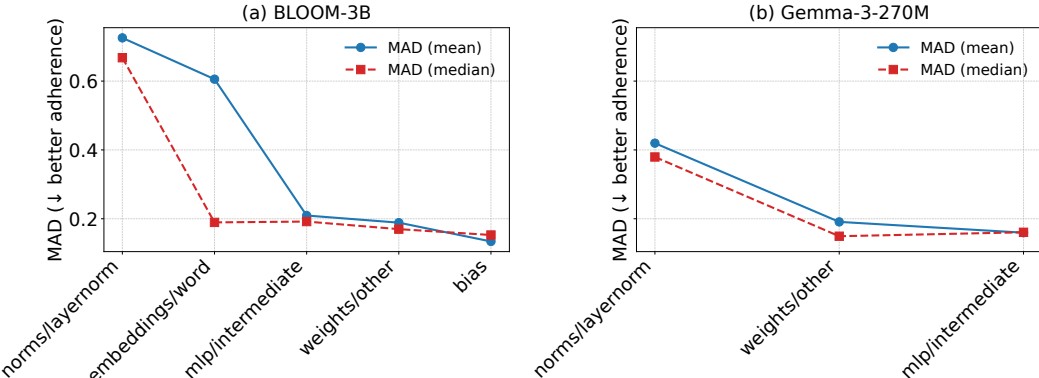

Figure 2: Comparison of Benford's Law non-compliance across weight families for two models. (a) **BLOOM-3B** shows consistently higher deviations, particularly in LayerNorm weights. (b) **Gemma-3-270M** exhibits overall lower deviations, although LayerNorm remains the dominant source of non-compliance. Lines denote the mean MAD per family, with markers indicating the median.

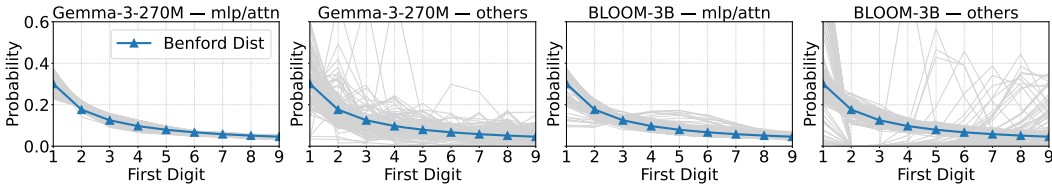

Figure 3: **Benford compliance across layer types.** Comparison of digit distributions between two models (Gemma-3-270M vs. BLOOM-3B). Attention and MLP layers exhibit closer adherence to Benford's law, while other layers diverge more strongly.

**Findings.** Our primary finding is a strong dichotomy based on layer functionality, as illustrated in Figure 2. We consistently observe (Figure 3) that weights from transformational 'nn.Linear' layers (e.g., in attention and feed-forward blocks) closely follow Benford's distribution. This provides strong motivation for a logarithmically-spaced quantizer. In contrast, 'nn.LayerNorm' weights systematically violate the law, with their values clustering around a single learned scalar (e.g., 0.35). We hypothesize these weights function not as transformations, but as learned damping factors to ensure network stability. This key finding motivates our selective quantization strategy, where 'LayerNorm' layers are excluded from quantization.

## 5.2 ABLATION STUDY: VALIDATING THE LOGARITHMIC GRID (RQ2)

**Setup.** Having established a potential link to logarithmic distributions in RQ1, we now validate the core design of Benford-Quant. We ask: is the Benford-inspired logarithmic spacing of quantization levels essential, or would any non-uniform grid suffice? To answer this, we conduct an ablation study on three transformer-based models comparing our proposed 'log-uniform' grid against a simpler 'linear non-uniform' baseline.

For this baseline, the positive quantization levels $\mathcal{L}^+_{\text{linear}}$ are spaced linearly within the normalized range $(0, 1]$, defined as:

$$\mathcal{L}^+_{\text{linear}} = \{k/N^+ \mid k \in \{1, \ldots, N^+\}\} \tag{10}$$

where $N^+ = 2^{B-1}$ is the number of positive levels. This creates a non-uniform grid where the absolute spacing between levels is constant, in contrast to our logarithmic grid where it is not. All other hyperparameters (4-bit, group size 128, selective quantization) are held constant between the two runs.

**Findings.** The results in Table 1 demonstrate the superiority of the logarithmic spacing. The 'log-uniform' grid yields a lower perplexity, confirming that allocating more precision to lower-magnitude

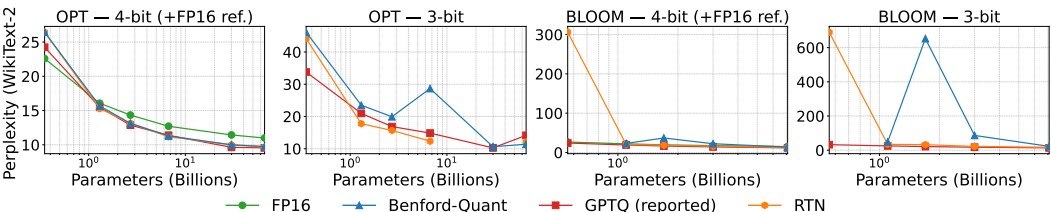

Figure 4: **Perplexity vs Parameters.** Results for OPT and BLOOM families across bit-widths. Our method (Benford) compared against reported GPTQ baselines. FP16 is shown as reference.

values—as predicted by Benford's Law—can boost performance. This supports the central hypothesis of our method. However, an exception is observed in the results for Llama2: the perplexity of the non-uniform linear distribution was lower than that of the log-uniform distribution. A plausible hypothesis to explain this situation is related to the quality of the data and/or the training process, given the direct correlation between the compliance of the weights with Benford's law and the model's generalization capability (Ott et al., 2025b).

Table 1: Ablation on level distributions for 4-bit quantization. **Takeaway:** The Benford-inspired *log-uniform* schedule is competitive and often better than a simple non-uniform *linear* baseline, although the best choice is mildly model-dependent (e.g., LLaMA2-7B).

| Model | Level Distribution | Perplexity WikiText2 $\downarrow$ |
|---|---|---|
| QWEN-7B-CHAT | Linear Non-Uniform | 9.15 |
| | **Log-Uniform (Ours)** | **9.12** |
| LLAMA2-7B | Linear Non-Uniform | **5.71** |
| | **Log-Uniform (Ours)** | 5.76 |
| GEMMA3-1B-IT | Linear Non-Uniform | 32.37 |
| | **Log-Uniform (Ours)** | **31.95** |
| OPT-66B | Linear Non-Uniform | 9.79 |
| | **Log-Uniform (Ours)** | **9.76** |

## 5.3 RQ3: COMPARISON WITH BASELINES

**Setup.** Finally, with our method's premise and design validated, we compare the full Benford-Quant algorithm (using selective quantization and log-uniform levels) against a strong uniform Round-to-Nearest (RTN) baseline. We evaluate both methods in 4-bit and 3-bit settings across all five model families.

**Results.** The results are summarized in Table 2. On the smaller Gemma-270M model, Benford-Quant provides a substantial improvement, reducing 4-bit perplexity from 38.91 (uniform) to 32.28. This confirms that for models with more "natural" weight distributions, our logarithmic grid is superior. On medium/large size models BENQ remains competitive but is sometimes surpassed by uniform RTN baseline. However, when analysing stability across bit quantities, our method has proven to be a good replacement for uniform RTN. We discuss this nuance in Section 6.

**Takeaway.** Across small language models (SLMs), Benford-Quant consistently improves perplexity over uniform RTN at 3–4 bits (e.g., Gemma-3-270M: 32.28 vs. 39.49 at 4b; Gemma-3-1B-it: 31.93 vs. 32.25 at 4b), supporting our hypothesis that SLM weight distributions benefit from a logarithmic allocation of precision. On mid-sized LLMs (e.g., OPT-1.3B, BLOOM-1b1/1b7), Benford-Quant remains competitive but does not universally dominate uniform RTN. We attribute this difference to a distributional shift in larger models (Section 5.1) and to implementation-level packing constraints at 3b (physically stored as 4b). We provide detailed per-layer analyses and ablations in the Appendix.

Table 2: **Perplexity (WikiText-2)** at low bit-widths. We report our results for `Uniform-RTN` and `Benford-Quant`, both with group size 8, alongside baselines from the literature: GPTQ (Frantar et al., 2022), SpQR and Qwen numbers (Jin et al., 2024), AFPQ (Zhang et al., 2023), AWQ (Lin et al., 2024), and GANQ (Zhao & Yuan, 2025). Lower is better. A dagger ($^{\dagger}$) denotes divergence; "–" = not available.

| Method | Bits | Small | | Medium | | Large | | |
| | | GEMMA-3 270M | OPT-1.3B | OPT-2.7B | QWEN-14B CHAT | OPT-30B | OPT-66B | QWEN-72B CHAT |
|---|---|---|---|---|---|---|---|---|
| RTN | 3 | 755.19 | 17.76 | 15.68 | $^{\dagger}$ | 10.89 | 12.08 | $^{\dagger}$ |
| | 4 | 38.91 | **15.30** | 13.13 | **7.50** | **9.99** | **9.68** | 17.22 |
| BENQ | 3 | 70.87 | 23.51 | 19.92 | 10.76 | 10.63 | 11.40 | 9.77 |
| | 4 | **32.28** | 15.67 | 13.14 | 7.90 | **9.99** | 9.76 | **7.02** |
| AFPQ | 4 | – | – | – | – | – | – | – |
| AWQ | 3 | – | 16.32 | 13.58 | – | 9.77 | – | – |
| GPTQ | 3 | – | – | 16.88 | 9.68 | 10.27 | 14.16 | – |
| | 4 | – | 15.47 | 12.87 | 7.35 | **9.63** | **9.55** | 6.37 |
| GANQ | 4 | – | **14.94** | 12.33 | – | – | – | – |
| SpQR | 3 | – | – | – | 7.31 | – | – | – |
| | 4 | – | – | – | **7.07** | – | – | **6.23** |

Our Uniform-RTN and Benford-Quant results use the same evaluation protocol but were run on a single H200. Note the strong stability of Benford-Quant at low bits, while Uniform-RTN diverges or underperforms.

Table 3: **Zero-shot LAMBADA accuracy** for representative models. Higher is better. Values for GPTQ are taken from the original papers (reported on A100), while Uniform-RTN and Benford-Quant are from our runs on H200. "–" = not available.

| Model | Model Accuracy | | | | |
| | QWEN-7B | QWEN3-14B | OPT-1.3B | OPT-2.7B | OPT-30B |
|---|---|---|---|---|---|
| Uniform-RTN (4bits) | **38.75** | **38.01** | **29.20** | 31.59 | 36.67 |
| Benford-Quant (4bits) | 32.56 | 35.52 | 29.05 | **32.00** | **36.77** |
| FP16 (16bits) | **38.36** | **38.36** | – | – | – |
| GPTQ (4bits,reported) | – | – | **56.45** | **62.97** | **69.12** |

Reported GPTQ numbers from Frantar et al. (2022). Our Uniform-RTN and Benford-Quant results use the same evaluation protocol.

## 6 DISCUSSION AND LIMITATIONS

The performance difference between SLMs and LLMs is a key finding. On SLMs such as Gemma-270M, Benford-Quant substantially improves perplexity, indicating that their weight distributions strongly benefit from a logarithmic grid. However, on larger models, the gains diminish and the uniform baseline occasionally outperforms. We hypothesize that this is due to over-parameterization: as models grow, their weight distributions flatten and diverge from the log-law structure, reducing the benefit of our method. An alternative hypothesis would be related to the training and quality of the data, as already pointed out by Ott et al. (2025a). This observation motivates further research into hybrid quantization strategies that adapt grid spacing dynamically according to empirical distributions.

Our method is currently data-free. While this is an advantage in terms of simplicity and generality, it also means we do not exploit activation statistics during calibration. Activation-aware quantization methods such as AWQ (Lin et al., 2024) or SmoothQuant (Xiao et al., 2023) could be complementary to our non-uniform weight quantization grid. Integrating Benford-Quant with these techniques is a promising avenue for improving robustness in LLM-scale deployments.

## 7 CONCLUSION

We introduced BENQ, a non-uniform weight quantizer whose codebook is derived from Benford's law. Empirically, the Benford prior fits *transformational* layers (attention/MLP linears) but not

Table 4: **Perplexity (C4)** (lower is better). Our results (`Uniform-RTN`, `Benford-Quant`) are compared against *reported* baselines: GPTQ and SpQR from Jin et al. (2024), AWQ from Zhang et al. (2023), and GANQ from Zhao & Yuan (2025). A dagger ($^\dagger$) denotes divergence; "—" = not available.

| Model | Bits | Uniform-RTN | Benford-Quant (ours) | Baselines (reported) | | |
|---|---|---|---|---|---|---|
| | | | | GPTQ | AWQ/SpQR | GANQ |
| LLAMA-2 7B | 4 | 11.15 | **11.13** | 11.45 | **11.05** (AWQ) | 11.25 |
| QWEN-14B-CHAT | 3 | $^\dagger$ | **22.60** | 14.59 | **10.92** (SpQR) | — |
| | 4 | **13.09** | 14.15 | 10.99 | **10.64** (SpQR) | — |

Values under "Baselines (reported)" are taken directly from the cited papers. Our runs use a single NVIDIA H200; the reported baselines typically use a single A100.

Table 5: **Runtime for full PTQ** on evaluated models. Our times are on a single NVIDIA H200 (weight-only PTQ; W4 shown). **GPTQ times (displayed in parenthesis)** are as reported by the original paper on a single NVIDIA A100. This highlights the order-of-magnitude efficiency gap, even when CPU quantization was needed. * = Quantized on CPU due to lack of GPU memory.

| Method | GEMMA-3-270M | GEMMA-3-1B-IT | QWEN-7B | QWEN3-14B | QWEN-7B-CHAT | QWEN-14B-CHAT |
|---|---|---|---|---|---|---|
| BENQ | 0.22s | 0.34s | 0.92s | 1.29s | 1.45s | 2.37s |
| RTN | 0.11s | 0.15s | 0.60s | 0.70s | 0.87s | 1.23s |
| | BLOOM-560M | BLOOM-1.1B | BLOOM-1.7B | BLOOM-3B | BLOOM-7.1B | QWEN-72B-CHAT |
| BENQ | 0.25s | 0.29s | 0.29s | 0.51s | 0.74s | 11,36m* |
| RTN | 0.11s | 0.13s | 0.15s (2.9m) | 0.21s (5.2m) | 0.38s (10m) | 4,77m* |
| | OPT-350M | OPT-1.3B | OPT-2.7B | OPT-6.7B | OPT-30B | OPT-66B |
| BENQ | 0.45s | 0.29s | 0.45s | 0.92s | 2.66s | 7,91m* |
| RTN | 0.22s | 0.17s | 0.25s | 0.55s | 1.57s (44.9m) | 3,79m* (1.6h) |

normalization or embedding layers, motivating a selective, layer-aware application that preserves accuracy while retaining efficiency.

Across scales, BENQ is a strong drop-in alternative to uniform RTN. On **small** models (Gemma-3 270M, OPT-1.3B), it consistently improves 3–4 bit perplexity over Uniform-RTN. On **medium** models (OPT-2.7B, Qwen-14B-Chat), BENQ is competitive with GPTQ on Qwen-14B. On **large** models (OPT-30B/66B, Qwen-72B-Chat) BENQ performs **very strongly**— **near GPTQ at 4 bits**. These trends align with our hypothesis: a logarithmic, weight-prior-aware grid reduces error where the mass of weights is concentrated at small magnitudes, complementing activation-aware preconditioning.

Because BENQ only replaces the level-generation step, it is *orthogonal* to methods that operate in different scopes (e.g., SmoothQuant's activation smoothing; AWQ's activation-driven scaling). This makes hybridization straightforward: AWQ/SmoothQuant can precondition activations, while BENQ supplies a data-free, non-uniform codebook for the weights.

## REPRODUCIBILITY STATEMENT

The code related to this work was submitted alongside this paper in the Supplementary Materials field. We also make available, in the Appendix, a description of the technology stack we used.

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

# A APPENDIX

## LLM USAGE

We used large language models as assistive tools for proofreading the manuscript and for code documentation or refactoring. All suggestions were reviewed and edited by the authors, who take full responsibility for the final content.

## A.1 QUANTIZATION/DEQUANTIZATION EXAMPLE

Figure 5 presents a numerical example of BENQ.

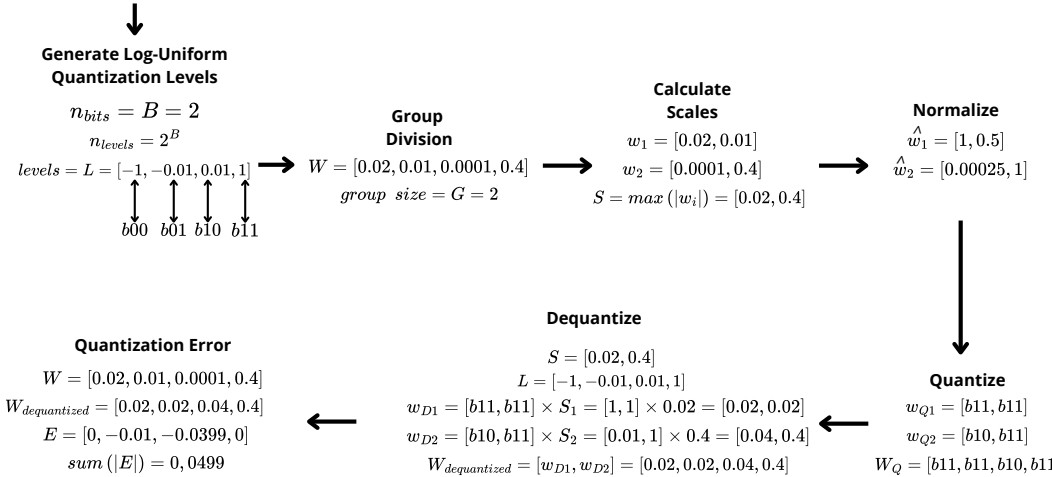

Figure 5: A numerical example of BENQ in a 2-bit scenario. Our main contribution resides on the log-uniform quantization levels.

## A.2 EXPERIMENT/IMPLEMENTATION DETAILS

This section provides additional information on how we conducted our experiments and implementation.

### A.2.1 TECHNOLOGY STACK

Our implementation used the following libraries/frameworks with the following versions: PyTorch 2.7.0; Numpy 2.1.3; Transformers 4.56.1; Datasets 4.1.0; Accelerate 1.10.1; Evaluate 0.4.5; Scikit Learn 1.7.2; Scipy 1.16.2; Matplotlib 3.10.6; and Seaborn 0.13.2. The operating system of the machine was Ubuntu 22.04.5 LTS and CUDA drivers were at version 13.0.

### A.2.2 Perplexity Evaluation

To ensure a fair comparison of results, we calculate perplexity the same as Frantar et al. (2022) (GPTQ authors), which in turn is the method defined by Radford et al. (2019). It consists of the following consecutive steps: **1.** Concatenate the validation dataset with two linebreaks as separators; **2.** Encode it with the model's tokenizer; **3.** Split it into non-overlapping segments of size 2048; **4.** Feed them to the model and collect the log-probabilities of each next token; and **5.** finally report perplexity as their exponentiated average.

### A.3 Additional Results

**Additional Analysis by Layer Families** To complement the main results, we analyzed the adherence of different layer families to Benford's Law across several models. Figures 6 and 7 summarize family-level deviations using the MAD score, as well as digit-wise signed deviations.

A consistent pattern emerges across models: **LayerNorm layers** are systematically the worst offenders, showing large deviations from Benford's expected distribution, while **MLP intermediate layers** tend to be the most Benford-like. Embeddings and bias terms exhibit intermediate behavior. These findings suggest that quantization strategies could benefit from selectively adapting to the family of the layer (e.g., using more conservative quantization for LayerNorm and embeddings, while applying more aggressive compression to MLP layers).

This family-level perspective highlights that distributional properties of weights are not uniform across the architecture, opening space for hybrid quantization heuristics that combine statistical laws with structural priors.

### A.4 Per-family Benford adherence (MAD)

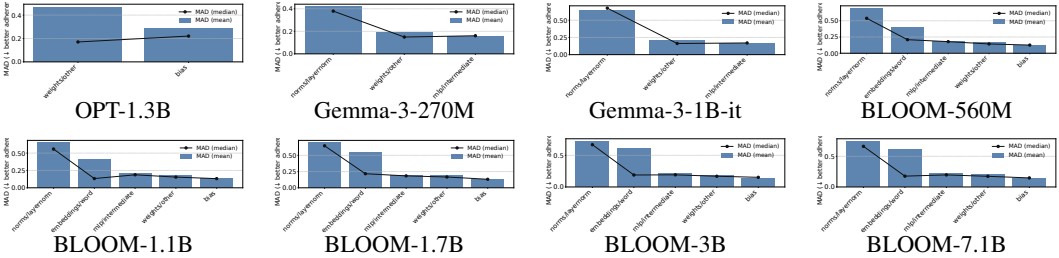

OPT-1.3B    Gemma-3-270M    Gemma-3-1B-it    BLOOM-560M

BLOOM-1.1B    BLOOM-1.7B    BLOOM-3B    BLOOM-7.1B

Figure 6: **Family-level Benford adherence (MAD; lower is better).** Bars show mean MAD per family; black dots trace the median. Normalization families (*LayerNorm*) consistently show the largest deviation from Benford; attention/MLP families show smaller MAD.

### A.5 Layer-wise Benford Adherence Analysis

To complement the family-level analysis presented in the main paper, we report a layer-wise view of Benford adherence, ordered by the Mean Absolute Deviation (MAD) score of each layer. Figures 8–14 show, for several model sizes and families (OPT, Gemma, BLOOM), how much each individual layer deviates from Benford's Law.

A consistent pattern emerges: only a small fraction of layers exhibit strong violations of Benford's Law, while the majority remain relatively well aligned. This results in a characteristic "long-tail" distribution: a short plateau of high-MAD layers (worst offenders), followed by a steep drop and then a flat region with near-zero deviation. Importantly, the worst cases are disproportionately large in magnitude compared to typical layers, often dominating the overall deviation score.

**Practical implications.** These results suggest that most of the quantization difficulties can be attributed to a handful of layers. From an engineering perspective, this means that selective strategies— such as allocating higher precision, applying per-layer calibration, or excluding a small subset of layers from aggressive quantization—may achieve most of the benefits of fine-grained quantization, without

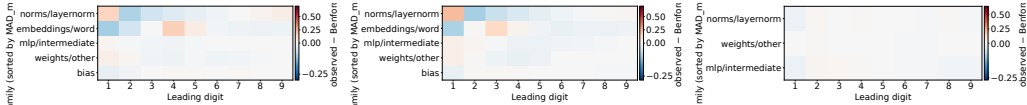

Figure 7: **Digit-wise signed deviation heatmaps. Models BLOOM 1.7B, BLOOM 1.1B and Gemma 3 270M respectively.** The plots reveal systematic over- and under-representation of leading digits across families. LayerNorm layers deviate the most, while MLP layers follow Benford closely.

increasing the cost across the entire model. Moreover, the elbow-shape visible in the distributions provides a natural thresholding criterion (either top-$k$ layers or MAD $> \tau$) for automated layer selection.

**Figures:**

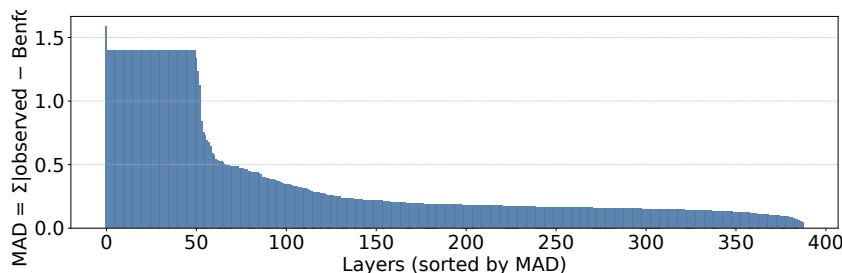

Figure 8: Layer-wise Benford MAD scores for OPT-1.3B. A small number of layers dominate the deviation.

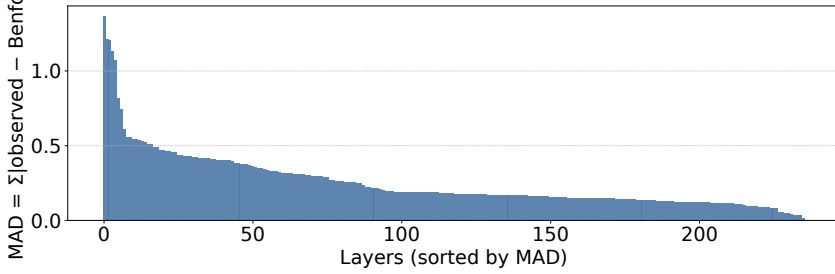

Figure 9: Layer-wise Benford MAD scores for Gemma-3 270M. Similar elbow-shaped pattern emerges, with a few high-MAD layers.

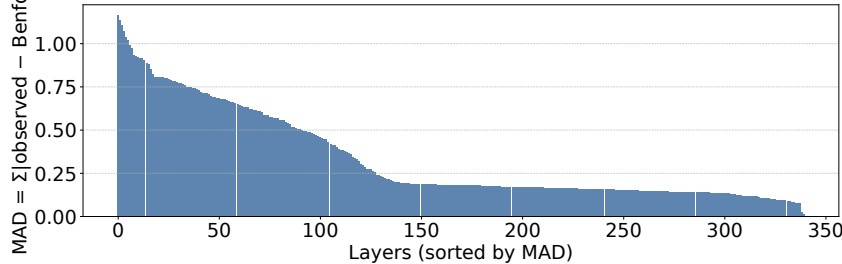

Figure 10: Layer-wise Benford MAD scores for Gemma-3 1B-it. Stronger deviations appear concentrated in a few layers.

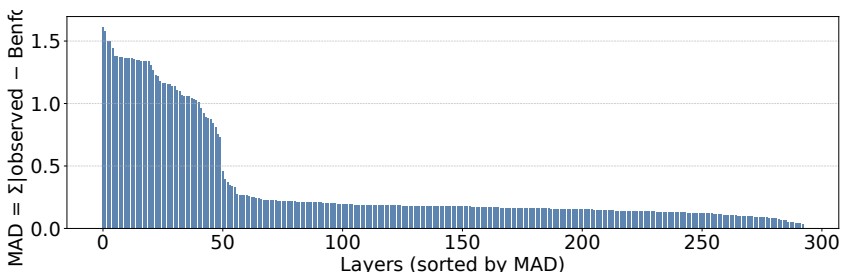

Figure 11: Layer-wise Benford MAD scores for BLOOM-560M. Most layers adhere well, with a few clear outliers.

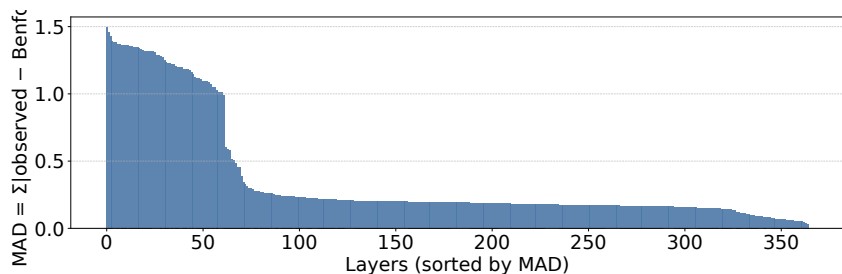

Figure 12: Layer-wise Benford MAD scores for BLOOM-3B. The elbow pattern is even sharper.

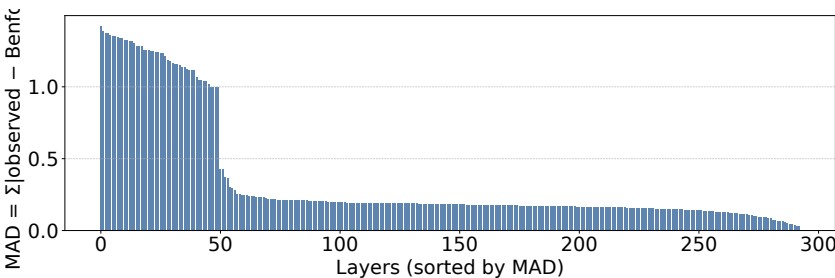

Figure 13: Layer-wise Benford MAD scores for BLOOM-1.7B. Again, a minority of layers account for most of the deviation.

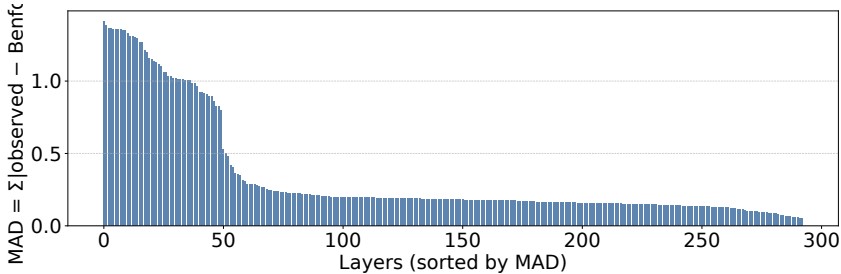

Figure 14: Layer-wise Benford MAD scores for BLOOM-1.1B. Consistent long-tail structure with few high-MAD outliers.

## A.6 ADDITIONAL RESULTS

Table 6: LLaMA-2-7B (HF): extra quantization results (appendix). Perplexities are on WikiText-2 (PPL) and C4 (PPL-C4). A dash (–) indicates not reported. Group size (G).

| Model | Method | Bits | G | PPL | PPL-C4 | Quant. Time (s) |
|---|---|---|---|---|---|---|
| Llama-2-7b-hf | benford-quant | 4 | 8 | 5.7583 | 11.1317 | 1.477 |
| Llama-2-7b-hf | benford-quant | 4 | 64 | 5.8754 | 11.5087 | 1.409 |
| Llama-2-7b-hf | benford-quant | 4 | 128 | 5.9186 | 11.6277 | 1.387 |
| Llama-2-7b-hf | benford-quant | 4 | 256 | 6.7802 | 14.6787 | 1.317 |
| Llama-2-7b-hf | uniform-rtn | 4 | 8 | 5.7550 | 11.1515 | 0.777 |
| Llama-2-7b-hf | uniform-rtn | 4 | 64 | 5.8871 | 11.5447 | 0.756 |
| Llama-2-7b-hf | uniform-rtn | 4 | 128 | 5.9465 | 11.7606 | 0.731 |
| Llama-2-7b-hf | uniform-rtn | 4 | 256 | 6.0631 | 12.1334 | 0.717 |

Table 7: Qwen family: extra quantization results (appendix). Perplexities are on WikiText-2 (PPL) and C4 (PPL-C4); LAMBADA is accuracy. A dash (–) indicates not reported. Very large PPL values denote divergence.

| Model | Method | Bits | G | PPL | PPL-C4 | LAMBADA | Quant. Time (s) |
|---|---|---|---|---|---|---|---|
| *Qwen-14B-Chat* | | | | | | | |
| Qwen-14B-Chat | fp16 | 0 | – | 7.1678 | 12.3494 | – | – |
| Qwen-14B-Chat | benford-quant | 2 | – | 28.6590 | 80.3259 | – | 2.393 |
| Qwen-14B-Chat | benford-quant | 3 | – | 10.7628 | 22.6016 | – | 2.374 |
| Qwen-14B-Chat | benford-quant | 4 | – | 7.9022 | 14.1532 | – | 2.367 |
| Qwen-14B-Chat | uniform-rtn | 2 | – | 8.6069e+08 | 2.8897e+08 | – | 1.263 |
| Qwen-14B-Chat | uniform-rtn | 3 | – | 1.2513e+07 | 5.0969e+06 | – | 1.218 |
| Qwen-14B-Chat | uniform-rtn | 4 | – | 7.4953 | 13.0932 | – | 1.232 |
| *Qwen-72B-Chat* | | | | | | | |
| Qwen-72B-Chat | benford-quant | 2 | – | 61.5665 | – | – | 629.195 |
| Qwen-72B-Chat | benford-quant | 3 | – | 9.7680 | – | – | 636.147 |
| Qwen-72B-Chat | benford-quant | 4 | – | 7.0215 | – | – | 681.663 |
| Qwen-72B-Chat | uniform-rtn | 2 | – | 1.0691e+16 | – | – | 280.460 |
| Qwen-72B-Chat | uniform-rtn | 3 | – | 4.0689e+07 | – | – | 272.021 |
| Qwen-72B-Chat | uniform-rtn | 4 | – | 17.2168 | – | – | 286.651 |
| *Qwen-7B* | | | | | | | |
| Qwen-7B | fp16 | 0 | – | 7.7116 | – | 0.3837 | – |
| Qwen-7B | benford-quant | 2 | – | 3029.3801 | – | 0.00019 | 1.142 |
| Qwen-7B | benford-quant | 3 | – | 20.1101 | – | 0.22763 | 1.130 |
| Qwen-7B | benford-quant | 4 | – | 9.1908 | – | 0.32564 | 0.924 |
| Qwen-7B | uniform-rtn | 2 | – | 1.2695e+09 | – | 0.00000 | 0.435 |
| Qwen-7B | uniform-rtn | 3 | – | 8.0331e+07 | – | 0.01669 | 0.474 |
| Qwen-7B | uniform-rtn | 4 | – | 8.2733 | – | 0.38754 | 0.602 |
| *Qwen-7B-Chat* | | | | | | | |
| Qwen-7B-Chat | fp16 | 0 | – | 8.8441 | 17.1559 | – | – |
| Qwen-7B-Chat | benford-quant | 2 | – | 382.5186 | 1248.8378 | – | 1.547 |
| Qwen-7B-Chat | benford-quant | 3 | – | 16.6403 | 43.7811 | – | 1.450 |
| Qwen-7B-Chat | benford-quant | 4 | – | 9.1198 | 18.1345 | – | 1.451 |
| Qwen-7B-Chat | uniform-rtn | 2 | – | 1.2658e+09 | 1.6446e+08 | – | 0.857 |
| Qwen-7B-Chat | uniform-rtn | 3 | – | 8.6900e+07 | 1.5832e+07 | – | 0.699 |
| Qwen-7B-Chat | uniform-rtn | 4 | – | 9.2380 | 18.0762 | – | 0.875 |
| *Qwen3-14B* | | | | | | | |
| Qwen3-14B | fp16 | 0 | – | 9.1166 | – | 0.38366 | – |
| Qwen3-14B | benford-quant | 2 | – | 48.6012 | – | 0.05492 | 1.469 |
| Qwen3-14B | benford-quant | 3 | – | 16.2297 | – | 0.20357 | 1.458 |
| Qwen3-14B | benford-quant | 4 | – | 10.3096 | – | 0.35319 | 1.292 |
| Qwen3-14B | uniform-rtn | 2 | – | – | – | 0.00000 | 0.763 |
| Qwen3-14B | uniform-rtn | 3 | – | 2.7310e+08 | – | 0.00000 | 0.706 |
| Qwen3-14B | uniform-rtn | 4 | – | 9.2654 | – | 0.38017 | 0.704 |

Table 8: BLOOM family: extra quantization results (appendix). Perplexities are on WikiText-2 (PPL). LAMBADA is accuracy (not reported here). "G" (group size); if absent we show "–".

| Model | Method | Bits | G | PPL | PPL-C4 | LAMBADA | Quant. Time (s) |
|---|---|---|---|---|---|---|---|
| *bloom-560m* | | | | | | | |
| bloom-560m | benford-quant | 2 | – | 9.2628e+20 | – | – | 0.224 |
| bloom-560m | benford-quant | 3 | – | 7.7121e+14 | – | – | 0.472 |
| bloom-560m | benford-quant | 4 | – | 44689.2422 | – | – | 0.249 |
| bloom-560m | none | 0 | – | 23.3777 | – | – | – |
| bloom-560m | uniform-rtn | 2 | – | 2.0271e+09 | – | – | 0.128 |
| bloom-560m | uniform-rtn | 3 | – | 690.2278 | – | – | 0.116 |
| bloom-560m | uniform-rtn | 4 | – | 306.0358 | – | – | 0.113 |
| *bloom-1b1* | | | | | | | |
| bloom-1b1 | benford-quant | 2 | – | 1619.4325 | – | – | 0.297 |
| bloom-1b1 | benford-quant | 3 | – | 48.4888 | – | – | 0.246 |
| bloom-1b1 | benford-quant | 4 | – | 23.1349 | – | – | 0.286 |
| bloom-1b1 | none | 0 | – | 18.4693 | – | – | – |
| bloom-1b1 | uniform-rtn | 2 | – | 52705.0742 | – | – | 0.134 |
| bloom-1b1 | uniform-rtn | 3 | – | 34.0214 | – | – | 0.157 |
| bloom-1b1 | uniform-rtn | 4 | – | 20.3793 | – | – | 0.133 |
| *bloom-1b7* | | | | | | | |
| bloom-1b7 | benford-quant | 2 | – | 8.1138e+06 | – | – | 0.294 |
| bloom-1b7 | benford-quant | 3 | – | 652.8582 | – | – | 0.338 |
| bloom-1b7 | benford-quant | 4 | – | 36.9484 | – | – | 0.293 |
| bloom-1b7 | none | 0 | – | 16.0000 | – | – | – |
| bloom-1b7 | uniform-rtn | 2 | – | 150178.5938 | – | – | 0.197 |
| bloom-1b7 | uniform-rtn | 3 | – | 32.2100 | – | – | 0.153 |
| bloom-1b7 | uniform-rtn | 4 | – | 18.6635 | – | – | 0.151 |
| *bloom-3b* | | | | | | | |
| bloom-3b | benford-quant | 2 | – | 17717.9648 | – | – | 0.437 |
| bloom-3b | benford-quant | 3 | – | 86.3477 | – | – | 1.055 |
| bloom-3b | benford-quant | 4 | – | 22.2959 | – | – | 0.515 |
| bloom-3b | uniform-rtn | 2 | – | 941.3247 | – | – | 0.219 |
| bloom-3b | uniform-rtn | 3 | – | 22.2131 | – | – | 6.227 |
| bloom-3b | uniform-rtn | 4 | – | 16.0303 | – | – | 0.209 |
| *bloom-7b1* | | | | | | | |
| bloom-7b1 | benford-quant | 2 | – | 113.3383 | – | – | 0.718 |
| bloom-7b1 | benford-quant | 3 | – | 22.2725 | – | – | 0.728 |
| bloom-7b1 | benford-quant | 4 | – | 13.8043 | – | – | 0.741 |
| bloom-7b1 | uniform-rtn | 2 | – | 231.0221 | – | – | 0.406 |
| bloom-7b1 | uniform-rtn | 3 | – | 16.7797 | – | – | 0.377 |
| bloom-7b1 | uniform-rtn | 4 | – | 12.6477 | – | – | 0.381 |

