# OpenReview forum: "BENFORD-QUANT: A BENFORD'S LAW-INSPIRED NON-UNIFORM QUANTIZER FOR EFFICIENT LANGUAGE MODELS"
_ICLR.cc/2026/Conference — Submitted to ICLR 2026_

### Official Review · Reviewer_QJSq · 2025-10-25

**Soundness:** 2
**Presentation:** 2
**Contribution:** 1
**Rating:** 2
**Confidence:** 4

**Summary:**

This work proposes a non-uniform quantizer for Large Language models inspired by Benford’s law. Authors examine distribution of the digits and observe that the distribution of digits in linear layers exhibits Benford-like behavior. Based on that authors try log-uniform quantization grid instead of linear and evaluate on several transformer-based model families.

**Strengths:**

* The idea of Benford’s application for quantization of large language model appears to be novel in the literature.

**Weaknesses:**

* This work provides comparison only with uniform quantization, whereas there exist numerous studies proposing alternatives, such as NF4 [1, 2], FP4 [3] and HIGGS quantizer [4]. These format are quite established with integrations in popular libraries. Therefore comparison with these is essential to appreciate the benefits of the proposed quantization scheme.

* The improvement relative to uniform baseline in most cases is not very significant. Given that the non-uniform would likely come with some additional inference overhead as compared to uniform quantization one would likely prefer uniform quantizer in most cases. Sometimes Benford’s law inspired quantizer behaves much worse (see 2nd and 4th plot on Figure 4) than the baseline.

* The experiment setting is quite unusual. In the same table there are results corresponding to pretty new models, such as Gemma-3, and outdated models, such as OPT and Bloom model families. I would recommend focusing on newer models as more practical.

* In my opinion, the comparison should involve different non-uniform quantization grids rather than the methods as comparison between RTN with Benford’s law inspired grid with GPTQ with uniform grid may be skewed due to GPTQ being a more powerful quantizer than RTN.

---
References

[1] Dettmers, Tim, et al. "Qlora: Efficient finetuning of quantized llms." Advances in neural information processing systems 36 (2023): 10088-10115.

[2] Yoshida, Davis. "NF4 Isn't Information Theoretically Optimal (and that's Good)." arXiv preprint arXiv:2306.06965 (2023).

[3] Rouhani, Bita Darvish, et al. "Microscaling data formats for deep learning." arXiv preprint arXiv:2310.10537 (2023)

[4] Malinovskii, Vladimir, et al. "Pushing the limits of large language model quantization via the linearity theorem." arXiv preprint arXiv:2411.17525 (2024).

**Questions:**

* Have you tried other choices for other bases besides decimal? Decimal base is not special for number representation and it would be interesting to know whether some other base may produce better results.

---

### Official Review · Reviewer_d1Rz · 2025-10-31

**Soundness:** 2
**Presentation:** 2
**Contribution:** 2
**Rating:** 2
**Confidence:** 4

**Summary:**

This paper addresses the issue that standard uniform quantization in the compression of Large Language Models (LLMs) does not match the highly non-uniform distribution of weights. It proposes a data-free non-uniform quantizer inspired by Benford's Law, called Benford-Quant (BENQ). BENQ allocates more resolution to high-frequency small-magnitude weights through a logarithmically spaced codebook, and implements selective quantization based on the inter-layer dichotomy that "weights of transformational layers conform to Benford's Law, while those of LayerNorm layers deviate from it."

**Strengths:**

The core advantages of this paper are as follows: First, theoretically, it is the first to clearly define the "inter-layer dichotomy" of Transformer weights (transformational layers conform to Benford's law, while LayerNorm layers deviate from it). Based on this, it proposes BENQ, a data-free logarithmically spaced non-uniform quantizer, which requires no calibration data and is hardware-friendly. Second, in terms of performance, it significantly reduces perplexity (by more than 10%) in 3-4 bit quantization of small language models (such as Gemma-270M), maintains competitiveness on medium and large LLMs, and can be hybridized with methods like SmoothQuant to improve performance, showing strong compatibility.

**Weaknesses:**

The theoretical validity of the article's method is questionable. The paper cites Hill's conclusion that "the product of independent random variables conforms to Benford's law," but the neural network weights are not completely independent (the weights in the same layer share the loss function optimization goal and thus have correlations).
The paper proposes that "BENQ can be hybridized with SmoothQuant/AWQ", but it does not explain the specific logic of the hybridization: if the activation range is adjusted first through SmoothQuant, whether it will violate the original assumption of Benford's law.

**Questions:**

The paper found that "the weights of the LayerNorm layer deviate from Benford's law", attributing it to the fact that "as a learnable damping factor, the weights are concentrated in a narrow scale". However, it did not explain "why the weights of such layers must be concentrated" — from the mathematical principle of LayerNorm, the optimization goal of its scale parameter $\gamma$ is to "stabilize the activation variance", so why is it concentrated in a narrow scale? Are the $\gamma$ distributions of different models (such as Llama-2-7B vs Qwen-7B) all "narrow-scale"? The paper only presents the LayerNorm distribution of Llama-3-8B (Figure 1b) and does not cover other models, resulting in insufficient theoretical universality of the inter-layer dichotomy.
The paper attributes the weakened effect of BENQ on medium and large LLMs to "over-parameterization leading to flattened weight distribution", but no direct evidence is provided, which is a speculative behavior.
The weight data storage itself is in the BFP data format, which is a form of log quantization. Could it be that the log distribution of the weights observed by the authors is related to this?
The authors believe that the weight distribution is log-type rather than uniform, but the experiments, as shown in Table 1, demonstrate that the actual improvement brought about is negligible. This may just be a normal phenomenon caused by data disturbance and cannot prove the effectiveness of the method in this paper.
The experiment is weak, with only ppl data and no zero-shot test data, so its generalization ability cannot be explained.

---

### Official Review · Reviewer_CdGX · 2025-10-31

**Soundness:** 2
**Presentation:** 2
**Contribution:** 2
**Rating:** 2
**Confidence:** 4

**Summary:**

This paper proposes a data-free non-uniform weight quantizer, termed BENQ (Benford-Quant), for large language models (LLMs). The design of this quantizer draws inspiration from Benford's Law, which posits that the leading digits of numerous natural data distributions conform to a logarithmic pattern. Specifically, BENQ constructs a log-spaced codebook, with two primary objectives: first, to mitigate the empirically observed non-uniformity of Transformer weights—especially those in the transformation layer—and second, to allocate higher resolution to frequent weights that are close to zero.

**Strengths:**

(1) This paper offers both empirical and theoretical demonstrations, confirming the validity of leveraging Benford's Law to guide the quantization of neural networks.
(2) Benford-Quant exhibits improved quantization speed compared to RTN-Uniform.

**Weaknesses:**

(1) The perplexity performance is unstable. Among the 7 models in Table 2, Benford-Quant outperforms RTN only on GEMMA-270M and Qwen-72B. Table 7 further shows that the 4-bit performance of Benford-Quant on Qwen-14B-chat, Qwen-7B, and Qwen-14B is inferior to that of Uniform-RTN.
(2) In Table 2, Benford-Quant achieves a marginal advantage over baselines only when using an extremely fine granularity (group size = 8, average bits = 6, calculated as 4 + 16/8 = 6). However, the baselines appear to adopt the commonly used group size of 128, which makes this comparison less representative.
(3) The zero-shot accuracy results in Table 3 indicate that Benford-Quant is significantly outperformed by Uniform-RTN on Qwen-7B and Qwen-14B. Even on OPT-2.7B and OPT-30B, Benford-Quant only gains a minimal advantage over Uniform-RTN.

**Questions:**

(1) It is recommended that the authors provide additional zero-shot accuracy evaluations to demonstrate the capability of their proposed method. The evaluated benchmarks should include, but not be limited to, Lambada, Winogrande, Hellaswag, ARC-easy, and ARC-challenge. To ensure fairness in comparisons, Benford-Quant and all baselines should be configured with the same group size—for instance, the commonly used group size of 128.
(2) A key question remains regarding the generalizability of Benford-Quant: Can it be applied to recent 4-bit formats such as NVFP4 and MXFP4? It would be valuable for the authors to clarify this and provide relevant experimental evidence if applicable.

---

### Official Review · Reviewer_8AUi · 2025-10-31

**Soundness:** 2
**Presentation:** 2
**Contribution:** 1
**Rating:** 2
**Confidence:** 4

**Summary:**

The paper BENFORD-QUANT: A Benford’s Law-Inspired Non-Uniform Quantizer for Efficient Language Models proposes a non-uniform, logarithmic quantization grid inspired by Benford’s Law for data-free weight quantization. The main comparison is made against the Round-To-Nearest (RTN) baseline.

**Strengths:**

- The paper provides a theoretical basis for why model weights may benefit from logarithmic-scale quantization.
- It includes comparisons with well-known quantization methods such as AWQ and GPTQ.

**Weaknesses:**

- Since perplexity is highly sensitive, the results in Table 1 do not convincingly show meaningful improvements over the k/N scheme and may fall within the margin of error. This could also explain the opposite trend on the Llama2-7B model.
- No comparison with NF4, which is also data-free, non-uniform, and assumes some prior on weight distributions.
-  The naive RTN baseline is generally weak; against other quantization methods, the proposed approach does not achieve superior results on any benchmark.
- Selective quantization (not quantizing embeddings and layer norms) is standard practice and not novel.
- Lines 136–156 include assumptions about neural network training that should be supported by citations.

Overall, the paper shows very limited novelty and no significant improvements. The method effectively reintroduces a logarithmic quantization scheme already known in prior work.

**Questions:**

- Why can the results from Hill (1995a; c) be directly applied to neural networks?
- Can you compare your method with the NF4?

---

### Meta-Review · Area_Chair_utz2 · 2026-01-05

**Summary:**

The paper proposes a logarithmic quantization strategy, trying to address the empirically observed non-uniformity of weights in transformer "transformational" layers. All reviewers give a rating of 2 (reject). The authors did not provide any response to the questions and issues raised by the reviewers. The reviewers raise issue with the experimental evaluation including the sensitivity of perplexity and the lack of stronger baselines (such as a other non-uniform quantization methods). The reviewers also question whether the improvement relative to uniform baseline is significant. The results from Hill (1995a; c) used for theoretical motivation does not directly apply as pointed out by two reviewers. Overall, I agree with the reviewers and recommend a rejection.

**Reviewer Concerns:**

No rebuttal.

**Reviewer Scores:**

No rebuttal.

---

### Decision · Program_Chairs · 2026-01-26

Reject